# Gestational diabetes in women with obesity; an analysis of clinical history and simple clinical/anthropometric measures

**Sara L. White** [1]*, **Dharmintra Pasupathy**[1¤a], **Shahina Begum**[1¤b], **Naveed Sattar**[2], **Scott M. Nelson**[3], **Paul Seed** [1], **Lucilla Poston**[1], on behalf of the UPBEAT consortium[¶]

**1** Department of Women and Children's Health, King's College London, London, United Kingdom, **2** Institute of Cardiovascular and Medical Sciences, University of Glasgow, Glasgow, United Kingdom, **3** School of Medicine, University of Glasgow, Level 2 New Lister Building, Glasgow Royal Infirmary, Glasgow, United Kingdom

¤a Current address: Reproduction and Perinatal Centre, Faculty of Medicine and Health, University of Sydney, Camperdown, Australia

¤b Current address: Washington-Singer Laboratories, Department of Psychology, College of Life and Environmental Sciences, University of Exeter, Exeter, United Kingdom

¶ Members of the UPBEAT Consortium are listed in the Acknowledgments.

* sara.white@kcl.ac.uk

**Data Availability Statement:** Due to the limitations of the consent provided by the patients in our study, and restrictions imposed by our funders we

## Abstract

### Aim

We assessed clinical risk factors, anthropometric measures of adiposity and weight gain to determine associations with development of GDM in a cohort of pregnant women with obesity.

### Methods

This was a secondary analysis of the UPBEAT trial of a complex lifestyle intervention in pregnant women with obesity (ISRCTN89971375). Clinical risk factors, and measures of adiposity and weight were assessed in the early 2nd trimester (mean 17 +0 weeks), and adiposity and weight repeated in the early 3rd trimester (mean 27 +5 weeks').

### Results

Of the 1117 women (median BMI 35.0 kg/m$^2$) with complete data, 25.8% (n = 304) developed GDM (IADPSG criteria, OGTT 24-28weeks). Using multivariable analysis, early clinical risk factors associated with later development of GDM included age (adj OR 1.06 per year; 95% CI 1.04–1.09), previous GDM (3.27; 1.34–7.93) and systolic blood pressure (per 10mmHg, 1.34; 1.18–1.53). Anthropometric measures positively associated with GDM included second trimester (mean 17+0 weeks) subscapular skinfold thickness, (per 5mm, 1.12; 1.05–1.21), and neck circumference (per cm, 1.11; 1.05–1.18). GDM was not associated with gestational weight gain, or changes in skinfolds thicknesses or circumferences between visits.

cannot make the data generally available. The UPBEAT Scientific Advisory Committee accepts applications for use of data from those who make a formal request, providing a description of the intended study on a research application form (UPBEAT RAF) available from Glen Nishku (glen.nishku@gstt.nhs.uk).

**Funding:** This study was supported by the National Institute of Health Research (RP-PG-0407-10452), Medical Research Council UK (MR/ L002477/1), Chief Scientist Office, Scottish Government Health Directorates (Edinburgh) (CZB/A/680), Biomedical Research Centre at Guys & St Thomas NHS Foundation Trust & King's College London and the NIHR Bristol Biomedical Research Centre, Tommy's Charity, UK (SC039280). SLW was supported by a fellowship from Diabetes UK (14/0004849). LP is an Emeritus National Institute for Health Research Senior Investigator (NI-SI-0512-10104). The funders had no role in study design, data collection and analysis, decision to publish, or preparation of the manuscript. www.nihr.ac.uk www.mrc.ac.uk www.cso.scot.nhs.uk www.tommys.org www.guysandstthomasbrc.nihr.ac.uk www.diabetes.org.uk.

**Competing interests:** SLW, DP, SB, SMN, PS and LP none. I have read the journal's policy and NS has consulted for Abbott Diagnostics, Amgen, AstraZeneca, Boehringer Ingelheim, Eli Lilly, MSD, Novo Nordisk, Pfizer, and Sanofi and his University has received funding from grants from Astrazeneca, Boehringer Ingelheim, Roche diagnostics. This does not alter our adherence to PLOS ONE policies on sharing data and materials.

## Conclusions

In this cohort of women with obesity, we confirmed clinical risk factors for GDM, (age, systolic blood pressure) previously identified in heterogeneous weight women but add to these indices of adiposity which may provide a discriminatory approach to GDM risk assessment in this group. This study also underscores the need to focus on modifiable factors pre-pregnancy as an opportunity for GDM prevention, as targeting gestational weight gain and adiposity during pregnancy is likely to be less effective.

## 1 Introduction

An estimated 14.6 million pregnant women were obese globally in 2014 [1] and data suggests that 21% of adult women will be obese worldwide by 2025 [2, 3]. As obesity is a strong risk factor for gestational diabetes (GDM) [4] the global rise in obesity has led to a concomitant increase in the prevalence of GDM.

Reported clinical and demographic factors associated with GDM in weight heterogeneous women include maternal demographic variables, family and social history, obstetric history, current pregnancy factors and clinical measurements [4–9]. Risk factors reported in national GDM guidelines include Body Mass Index (BMI), older age, parity, previous GDM, family history of diabetes and previous macrosomia [10–13]. The utility of these factors to define GDM risk amongst women with obesity, however, remains unclear.

BMI has been the focal assessment of GDM risk in weight heterogeneous women, with relatively little attention to more direct measures of adiposity such as skinfold thicknesses, and limb and waist circumferences which may better reflect pathological distributions of adipose tissue [14]. In women with obesity these measures when evaluated early in pregnancy, may add increased granularity to maternal GDM risk assessment which BMI does not provide. Gestational weight gain (GWG) is associated with the development of GDM amongst weight heterogeneous women, with one meta-analysis quoting an unadjusted odds ratio (OR) of 1.40 (95% CI 1.21–1.61) for the association between excessive GWG and GDM, with no evidence of interaction with maternal pre-pregnancy BMI [15]. Despite this evidence, interventions designed to improve lifestyle or reduce GWG in women with obesity, have not translated to a reduction in GDM [16–18].

Using maternal data from the UPBEAT randomised controlled trial (RCT) [16], this study has addressed the association of maternal clinical and demographic factors and anthropometric measures, assessed in the early second, and third trimester, with the development of GDM in pregnant women with obesity. We also addressed the relationship between gestational change in modifiable maternal factors such as weight gain and adiposity, and the risk of GDM.

## 2 Materials and methods

### 2.1 Study design and participants

This prospective cohort study was a secondary analysis utilising data from the UPBEAT Trial (ISRCTN 89971375). UPBEAT was a multicentre RCT of a complex dietary and physical activity intervention in pregnant women with obesity [16]. Women with a pre-existing diagnosis of essential hypertension, diabetes, coeliac disease, thyroid disease, renal disease, systemic lupus erythematosus, antiphospholipid syndrome, sickle-cell disease, thalassaemia, current psychosis, or a current prescription of metformin were excluded. The UPBEAT trial, which was

undertaken between 2009 and 2014, consisted of 1555 recruited women (out of 8820 assessed for inclusion) who were >16 years of age, had a BMI $\geq$ 30kg/m$^2$ and a singleton pregnancy. Women were randomised between 15$^{+0}$ and 18$^{+6}$ weeks' gestation to either a behavioural intervention superimposed on standard antenatal care or standard antenatal care. The primary outcomes of the trial, GDM and delivery of a large for gestational age (LGA) infant were no different between intervention and standard care arms. The intervention was associated with lower GWG and reduced skinfold thicknesses. All aspects of the trial, including the analyses of the present study were approved by the NHS Research Ethics Committee (UK Integrated Research Application System; reference 09/H0802/5) and all participants, including women aged 16 and 17 using Fraser guidelines, provided informed written consent [16].

For the purpose of this study, analysis was undertaken utilising women with available Oral Glucose Tolerance Test (OGTT) results who had complete data at study visits 1 and 2 (15–18$^{+6}$ and 27–28$^{+6}$ gestational weeks', respectively). This was followed by two sensitivity analyses; the first using an imputed dataset constructed by chained equations using auxiliary variables related to demographic and clinical variables and ten imputed datasets, representing all women with available OGTT data ($n$ = 1303); and the second, a restricted dataset with the removal of outliers. Outliers were defined either due to a measure falling $\geq$ 4SD from the variable mean, or if the variable was not measured within the pre-identified gestational windows.

This study is reported as per the Strengthening the Reporting of Observational Studies in Epidemiology (STROBE) guideline.

## 2.2 Maternal clinical factors

The selection of maternal clinical factors was based on *a priori* plausible association with GDM development and included socio-demographic factors and obstetric and family history, recorded at the first study visit. Clinical and anthropometric measures including maternal blood pressure, weight, height, circumferences and skinfold thicknesses (reflecting diverse patterns of adipose distribution including subcutaneous and visceral adiposity), were measured at both visits. Midwives underwent prior training in measurement methods. Circumferences were recorded to the nearest millimetre and skinfolds were measured in triplicate using Harpenden skinfold callipers.

GDM was defined utilising International Association of Diabetes and Pregnancy Study Groups (IADPSG) criteria from an OGTT carried out at the second study visit. A positive diagnosis comprised of one or more of the following; fasting glucose $\geq$ 5.1 mmol/l, 1hr glucose $\geq$ 10.0 mmol/l, 2hr glucose $\geq$ 8.5 mmol/l; 75g glucose load [19]. The trial protocol specified inclusion of OGTTs between 27 and 28$^{+6}$ gestational weeks', however a clinically pragmatic approach was adopted for this study with inclusion of OGTTs undertaken between 23$^{+0}$ and 32$^{+6}$ (mean 27$^{+5}$).

## 2.3 Statistical analysis

Statistical analysis was performed using Stata software, version 14.0 (StataCorp LP, College Station, Texas).

To address the aims of this study, the UPBEAT RCT was treated as a cohort study as the primary outcomes (GDM and LGA infants) did not differ between control and intervention groups.

Ratios of maternal measures (e.g. waist:hip) were constructed for inclusion as there is evidence these better reflect insulin resistance or diabetes risk than simple measures alone [20, 21]. Clinical and anthropometric measures were treated as continuous variables, and were assessed for normality, linearity and variation with gestational age. Maternal clinical

characteristics and anthropometric measures were summarised using means and percentages as appropriate. Summary statistics between those who developed GDM and those who did not were compared using either Student's *t*-test or Mann-Whitney tests for continuous data, or $\chi^2$ tests for categorical data as appropriate. Characteristics and measures found to be significantly associated with GDM at this stage were taken forward for inclusion in the regression models. Selection bias was assessed by comparing maternal characteristics between the complete dataset ($n = 1177$) and an imputed dataset of the study cohort ($n = 1303$).

Association between maternal clinical factors (second and third trimester) and GDM in women with obesity was assessed using logistic regression (unadjusted and adjusted analyses). Multiple regression models included clinical unmodifiable factors (recorded at visit 1, e.g. history of previous GDM) and modifiable factors (e.g. measures of adiposity) measured at each visit, to identify risk factors for GDM at each time point. Rate of change (weight and adiposity) between visits 1 and 2 (pre management for GDM) was also explored between groups using both unadjusted and adjusted analyses. As factors for analysis were restricted in number and chosen *a priori*, a threshold of $p < 0.05$ was utilised for significance testing.

To prevent the inclusion of highly correlated/colinear factors (e.g. skinfold measures) in multiple regression modelling, variables likely to be related were grouped *a priori* (S1 Table in S1 File). Correlation matrices were constructed (Stata; *pwcorr*) utilising a pre-identified threshold for correlation of $r^2 > 0.3$. Of those found to be correlated, the factor with the strongest association with the development of GDM from that group, when tested individually and based on z score, was selected for inclusion in subsequent modelling. Two models were constructed utilising either distinct simple measures e.g. separate skinfold thicknesses or circumferences, or summed skinfold thicknesses and ratios.

## 2.4 Selection of confounders

To identify factors associated with the development of GDM, measures selected in the procedure above were included in multivariable models with additional adjustment for maternal unmodifiable factors found to be associated with GDM (age, previous GDM and family history of Type 2 diabetes; T2DM). Ethnicity and parity were included as confounders, being known risk factors for GDM in weight heterogeneous women. At the second visit, randomisation arm allocation was included as an adjustment.

Analysis of rate of change between visits was adjusted for age, ethnicity, parity, previous GDM and family history of T2DM as well as randomisation arm allocation where further analysis was appropriate.

To test the hypothesis that risk associated with gestational weight gain interacts with maternal BMI, BMI was categorised into 3 WHO obesity groups, (30–34.9 kg/m$^2$, 35.0–39.9 kg/m$^2$, $\geq$ 40 kg/m$^2$) and relationships with rate of change of gestational weight gain evaluated.

## 2.5 Sensitivity analysis

A sensitivity analysis was carried out, in which individuals within the cohort were excluded if considered outliers, either due to a measure falling $\geq$ 4SD from the variable mean or if the variable was not measured within the pre-identified gestational windows (visit 1 restricted to 15–17$^{+6}$ weeks' and visit 2 restricted to 27$^{+0}$–28$^{+6}$ weeks').

## 3 Results

Of the 1555 participants in the UPBEAT trial, 1303 women underwent interpretable oral glucose tolerance testing (median BMI 35 kg/m$^2$). 1177 women (median BMI 35 kg/m$^2$, IQR 32.8–38.5) had complete data at both study visits. Of these, 304 (25.8%) developed GDM.

Mean gestational age (GA) at the first and second study visits were 17.1 weeks (SD 1.1) and 27.8 weeks (SD 0.7) respectively. Summary statistics for all maternal characteristics and measurements at each visit are shown in Tables 1–3.

When comparing women with complete clinical data (n = 1177) with the imputed dataset (*n* = 1303), there were no significant differences in any maternal variable, either clinical history or measured at either visit (S2–S4 Tables in S1 File).

## 3.1 Associations between maternal exposures and GDM at visits 1 and 2

Maternal summary statistics are shown in Tables 1–3, and unadjusted and adjusted analyses (logistic regression) at visits 1 and 2 in Tables 4 and 5, respectively.

Women who developed GDM were older, more likely to have had GDM in a previous pregnancy, and to have a first-degree family member with T2DM, than those who did not develop GDM. There were no differences in risk due to ethnicity, socio economic status as assessed

**Table 1. Maternal characteristics by GDM status.**

| Maternal factors | Complete-case dataset *n* = 1177 | |
| --- | --- | --- |
| | **No GDM** | **GDM** |
| | **(*n* = 873)** | **(*n* = 304, 25.8%)** |
| | **Mean (SD) or n (%)** | **Mean (SD) or n (%)** |
| **Age (years)** | **30.3 (5.5)** | **32 (4.9)*** |
| **Ethnicity** | | |
| African | 131 (15) | 56 (18.4) |
| African Caribbean | 70 (8) | 22 (7.2) |
| South Asian | 48 (5.5) | 19 (6.3) |
| European | 562 (64.4) | 182 (59.9) |
| Other | 62 (7.1) | 25 (8.2) |
| **Adjusted English & Scottish IMD** | | |
| least deprived | 209 (23.9) | 63 (20.7) |
| Intermediate | 315 (36.1) | 100 (32.9) |
| most deprived | 349 (40) | 141 (46.4) |
| **Parity** | | |
| Nulliparous | 395 (45.2) | 128 (42.1) |
| **Previous GDM** | | |
| No previous | 467 (53.5) | 163 (53.6) |
| Previous | **11 (1.3)** | **13 (4.3)** |
| Nulliparous | 395 (45.2) | 128 (42.1) |
| **PCOS** | 80 (9.2) | 35 (11.5) |
| **Current Smoking** | 53 (6.1) | 24 (7.9) |
| **Family History** | | |
| T2DM | **187 (21.4)** | **96 (31.6)*** |
| GDM | 31 (3.6) | 15 (4.9) |
| IHD | 128 (14.7) | 56 (18.4) |
| HTN | 392 (44.9) | 146 (48) |
| **Randomisation** | | |
| Intervention arm | 428 (49) | 150 (49.3) |

**Bold** *p* value <0.05, **bold*** <0.001 (*p* value from *t*-test or $\chi^2$ as appropriate). GDM gestational diabetes, SD standard deviation, IMD Index of multiple deprivation, PCOS polycystic ovarian syndrome, T2DM type 2 diabetes, IHD ischaemic heart disease, HTN hypertension

**Table 2. Maternal clinical and anthropometric measures in early second trimester (mean $17^{+0}$ weeks gestation) by GDM status (diagnosis at mean $27^{+5}$ weeks' gestation).**

| | Complete-case dataset $n$ = 1177 | |
|---|---|---|
| **Variable measured** | **No GDM** | **GDM** |
| | (**$n$ = 873**) | (**$n$ = 304**) |
| | **Mean (SD) or median (IQR)** | **Mean (SD) or median (IQR)** |
| Systolic BP (mmHg) | **116.8 (10.7)** | **120.9 (10.9)**\* |
| Diastolic BP (mmHg) | **71.5 (7.6)** | **74.3 (8.1)**\* |
| Weight (Kg) | **97.1 (13.8)** | **99.5 (16.4)** |
| Height (cm) | **164.3 (6.7)** | **163.5 (7.1)** |
| BMI (kg/m$^2$) | **34.7 (32.7–38.1)** | **36 (33.1–39.7)**\* |
| **Circumferences** | | |
| Waist (cm) | **106.3 (9.9)** | **110.3 (10)**\* |
| Thigh (cm) | 68.4 (6.3) | 68.7 (7.6) |
| Wrist (mm) | 172.2 (14.1) | 173.5 (13.6) |
| Mid-arm (cm) | **36.5 (3.9)** | **37.6 (4.1)**\* |
| Neck (cm) | **36.3 (2.4)** | **37.3 (2.6)**\* |
| Hip (cm) | 122.4 (9.7) | 123.6 (11.4) |
| **Skin folds (mean, mm)** | | |
| Biceps | **21.3 (7.5)** | **23.5 (8.4)**\* |
| Triceps | **32.4 (8.7)** | **34.5 (9.6)**\* |
| Subscapular | **34.4 (9.5)** | **38.2 (10.9)**\* |
| Suprailiac | **31.4 (11)** | **34.5 (10.8)**\* |
| Sum of skinfolds | **119.4 (25.7)** | **130.7 (29)**\* |
| **Ratios** | | |
| Waist:height | **0.65 (0.06)** | **0.68 (0.06)**\* |
| Waist:thigh | **1.56 (0.18)** | **1.62 (0.22)**\* |
| Waist:hip | **0.87 (0.07)** | **0.89 (0.07)**\* |
| Neck:thigh | **0.53 (0.05)** | **0.55 (0.07)**\* |
| GA at visit 1, weeks | 17.1 (1.1) | 17.1 (1.0) |

**Bold** $p$ value <0.05, **bold**\* $p$ value <0.001 ($p$ value from $t$-test or Mann Whitney as appropriate). GDM gestational diabetes, SD standard deviation, IQR interquartile range, BP blood pressure, GA gestational age

using the index of multiple deprivation (IMD), or parity (either nulliparous women or multiparous women who had not had previous GDM) (Table 1). The incidence of PCOS and smoking were also not different between GDM and non-GDM women (Table 1). Women who developed GDM were shorter in height and had a higher BMI at visit 1. Multiple clinical and anthropometric variables measured either in the early second or third trimester were different in women who did and did not develop GDM. The diastolic and systolic blood pressures were higher, as were the waist, mid-arm and neck maternal circumferences, as well as all measured skin fold thicknesses (biceps, triceps, suprailiac and subscapular, sum of skinfold thicknesses) and the waist:height, waist:thigh, waist:hip and neck:thigh ratios (Tables 2 and 3).

At both time points, the variables found to be associated with GDM in unadjusted analysis were then combined in a single model. In this analysis, older age and GDM in a previous pregnancy remained significant when adjusted for other factors (family history of T2DM, parity, ethnicity and other positive anthropometric measurements, simple or ratio) at both time points. Systolic blood pressure remained associated in all models at both time points. In model 1 (simple measures), subscapular skinfold thickness, neck (visit 1) and waist (visit 2)

**Table 3. Maternal clinical and anthropometric measures in early third trimester (mean 27$^{+5}$ weeks' gestation) by GDM status diagnosed at same gestation.**

| Variable measured | Complete-case dataset $n$ = 1177 | |
|---|---|---|
| | No GDM | GDM |
| | ($n$ = 873) | ($n$ = 304) |
| | Mean (SD) | Mean (SD) |
| Systolic BP (mmHg) | **117.5 (10.6)** | **120.8 (10.9)**\* |
| Diastolic BP (mmHg) | **72.4 (7.9)** | **74.1 (7.9)** |
| Weight (Kg) | **101 (13.9)** | **103.3 (16.6)** |
| **Circumferences** | | |
| Waist (cm) | **112.2 (9.1)** | **116.1 (9.4)**\* |
| Thigh (cm) | 68.8 (6.4) | 69.2 (7.6) |
| Wrist (mm) | 173.3 (13.6) | 174.1 (13.8) |
| Mid-arm (cm) | **36.4 (3.9)** | **37.6 (4.1)**\* |
| Neck (cm) | **36.3 (2.3)** | **37.3 (2.6)**\* |
| Hip (cm) | 124.2 (9.8) | 125.1 (11.6) |
| **Skin folds (mean, mm)** | | |
| Biceps | **21.4 (7.4)** | **22.9 (7.1)** |
| Triceps | **32.8 (8.2)** | **35.1 (9.1)**\* |
| Subscapular | **35.8 (8.9)** | **39.4 (10.3)**\* |
| Suprailiac | **32.3 (11)** | **35.4 (10.7)**\* |
| Sum of skinfolds | **122.3 (25.4)** | **132.8 (27.9)**\* |
| **Ratios** | | |
| Waist:height | **0.68 (0.05)** | **0.71 (0.06)**\* |
| Waist:thigh | **1.64 (0.17)** | **1.69 (0.17)**\* |
| Waist:hip | **0.91 (0.07)** | **0.93 (0.07)**\* |
| Neck:thigh | **0.53 (0.05)** | **0.54 (0.06)**\* |
| GA at visit 2 (OGTT), weeks | 27.8 (0.7) | 27.8 (0.7) |

**Bold** $p$ value <0.05, **bold**\* $p$ value <0.001 ($p$-value from $t$-test). GDM gestational diabetes, SD standard deviation, BP blood pressure, GA gestational age, OGTT oral glucose tolerance test

circumference were associated with GDM. Similarly, in model 2 (summed or calculated measures), sum of skinfold thicknesses, waist:height and neck:thigh ratios remained associated in adjusted analyses at both visits (Tables 4 and 5).

## 3.2 Sensitivity analyses

The same associations were found in multiple regression analyses using the imputed dataset ($n$ = 1303) (S5, S6 Tables in S1 File), with minimal differences in the complete dataset after removal of potential outliers (S7, S8 Tables in S1 File). With regards to the loss of association between previous GDM and the development of GDM, this likely reflects reduced power in the smaller dataset utilised for sensitivity analysis, particularly at visit 1, where 17 women with previous GDM were included in the analysis (compared to 24 in the whole cohort).

## 3.3 Associations between rate of change in maternal variables and GDM

Summary statistics and unadjusted analysis of rate change in anthropometric measures and BP are shown in S9 Table in S1 File. There was no difference in rate of change of any anthropometric measure between women who developed GDM and those who did not. There was an

**Table 4. Clinical history, and clinical and anthropometric variables measured at 17$^{+0}$ weeks' (mean) gestation associated with later development of GDM; unadjusted and multiple regression analyses.**

| $n$ = 1177 | | Unadjusted analysis | Adjusted analysis | Adjusted analysis |
|---|---|---|---|---|
| | | | Model 1^ | Model 2 $^>$ |
| | | OR (95% CI) | OR (95% CI) | OR (95% CI) |
| Age (years) | | **1.06 (1.04–1.09)*** | **1.06 (1.04–1.09)*** | **1.06 (1.03–1.09)*** |
| Previous GDM | No | 1 (ref) | 1 (ref) | 1 (ref) |
| | Yes | **3.39 (1.49–7.71)** | **3.27 (1.34–7.93)** | **3.11 (1.27–7.61)** |
| | Nulliparous | 0.93 (0.71–1.21) | 1.06 (0.79–1.42) | 1.15 (0.85–1.55) |
| 1$^{st}$ degree T2DM | Yes | **1.69 (1.27–2.26)*** | 1.37 (1.00–1.88) | 1.34 (0.85–1.55) |
| BMI | (kg/m$^2$) | **1.06 (1.03–1.09)*** | ~ | ~ |
| Weight | (kg) | **1.01 (1.00–1.02)** | / | / |
| Systolic BP | (per 10 mmHg) | **1.42 (1.25–1.60)*** | **1.34 (1.18–1.53)*** | **1.36 (1.19–1.55)*** |
| Diastolic BP | (per 10 mmHg) | **1.59 (1.34–1.88)*** | ~ | ~ |
| Skinfolds mean | Biceps | **1.19 (1.10–1.29)*** | ~ | / |
| (per 5mm) | Triceps | **1.13 (1.06–1.22)** | ~ | / |
| | Subscapular | **1.20 (1.13–1.29)*** | **1.12 (1.05–1.21)** | / |
| | Suprailiac | **1.14 (1.07–1.21)*** | ~ | / |
| | Sum of all | **1.08 (1.05–1.11)*** | / | **1.05 (1.02–1.08)** |
| Circumferences | Neck | **1.17 (1.11–1.24)*** | **1.11 (1.05–1.18)** | / |
| (cm) | Waist | **1.04 (1.03–1.05)*** | ~ | / |
| | Mid arm | **1.07 (1.04–1.10)*** | ~ | / |
| Ratios | Waist:hip | **1.63 (1.34–1.97)*** | / | ~ |
| Per 0.1 increase | Waist:thigh | **1.17 (1.09–1.26)*** | / | ~ |
| | Neck:thigh | **1.51 (1.21–1.88)*** | / | **1.58 (1.24–2.00)*** |
| | Waist:height | **2.05 (1.65–2.55)*** | / | **1.57 (1.23–2.01)*** |

**Bold** $p$ value <0.05, **bold*** $p$ value <0.001 ($p$ value from logistic regression)

^Model 1 (simple measures) adjusted for positive simple baseline variables selected after correlation analysis, plus ethnicity and parity

$^>$ Model 2 (ratios/summed values) adjusted for positive ratio baseline variables selected after correlation analysis, plus ethnicity and parity

/ not included *a priori*

~ not included after correlation testing

BP blood pressure, OR odds ratio, CI confidence interval

association with diastolic blood pressure change, however this was not evident on adjusted regression testing (S10 Table in S1 File) after adjustment for maternal age. Similarly, using the imputed dataset, there was no evidence of association with change in any measured variables, although change in mid-arm circumference was associated with GDM in the sensitivity analysis (S11-S13 Tables in S1 File).

To test the hypothesis that risk associated with weight gain might interact with maternal BMI, the association between gestational weight gain and GDM was stratified by BMI WHO category (S14, S15 Tables in S1 File) but no association was found in any group.

## 4 Discussion

This study explored the relationships between clinical history, as well as clinical and anthropometric measures recorded in the second and third trimesters, and the development of GDM in women with obesity. We also determined whether the rate of change in the anthropometric measures was associated with GDM.

**Table 5. Clinical history, and clinical and anthropometric variables measured at 27$^{+5}$ weeks' (mean) gestation associated with GDM diagnosis; unadjusted and multiple regression analyses.**

| n = 1177 | | Unadjusted analysis | Adjusted analysis | Adjusted analysis |
|---|---|---|---|---|
| | | | Model 1^ | Model 2 > |
| | | OR (95% CI) | OR (95% CI) | OR (95% CI) |
| Age (years) | | **1.06 (1.04–1.09)*** | **1.07 (1.04–1.10)*** | **1.07 (1.04–1.10)*** |
| Previous GDM | No | 1 (ref) | 1 (ref) | 1 (ref) |
| | Yes | **3.39 (1.49–7.71)** | **2.68 (1.12–6.39)** | **2.53 (1.04–6.14)** |
| | Nulliparous | 0.93 (0.71–1.21) | 1.03 (0.77–1.39) | 1.11 (0.82–1.49) |
| 1$^{st}$ degree T2DM | Yes | **1.69 (1.27–2.26)*** | **1.46 (1.07–2.00)** | 1.37 (0.99–1.88) |
| Weight | (kg) | **1.01 (1.00–1.02)** | ~ | ~ |
| Systolic BP | (per 10 mmHg) | **1.32 (1.17–1.49)*** | **1.29 (1.13–1.48)*** | **1.31 (1.14–1.50)*** |
| Diastolic BP | (per 10 mmHg) | **1.32 (1.12–1.56)** | ~ | / |
| Skinfolds | Biceps | **1.14 (1.05–1.24)** | ~ | / |
| (mean, per 5mm) | Triceps | **1.17 (1.08–1.26)*** | ~ | / |
| | Subscapular | **1.22 (1.14–1.31)*** | **1.14 (1.05–1.23)** | / |
| | Suprailiac | **1.13 (1.07–1.20)*** | ~ | / |
| | Sum of all | **1.08 (1.05–1.10)*** | / | **1.05 (1.02–1.08)** |
| Circumferences | Neck | **1.18 (1.11–1.24)*** | ~ | / |
| (cm) | Waist | **1.04 (1.03–1.06)*** | **1.03 (1.01–1.05)*** | / |
| | Mid arm | **1.08 (1.04–1.11)*** | ~ | / |
| Ratios | Waist:hip | **1.71 (1.41–2.07)*** | / | ~ |
| Per 0.1 increase | Waist:thigh | **1.19 (1.10–1.29)*** | / | ~ |
| | Neck:thigh | **1.44 (1.14–1.82)** | / | **1.52 (1.18–1.95)** |
| | Waist:height | **2.32 (1.83–2.93)*** | / | **1.87 (1.43–2.45)*** |

**Bold** p value <0.05, **bold***  p value <0.001 (p value from logistic regression)

^ Model 1 (simple measures) adjusted for positive simple time point 2 variables selected after correlation analysis plus age, ethnicity and parity, previous gestational diabetes (GDM,) family history T2DM, and intervention group

> Model 2 (ratios/summed measures) adjusted for positive ratio time point 2 variables selected after correlation analysis plus age, ethnicity, parity, previous GDM, family history T2DM, and intervention group

/ not included *a priori*

~ not included after correlation testing. BP blood pressure

Of the factors chosen *a priori* due to a putative or known association with GDM in weight heterogeneous women, many were confirmed amongst women with obesity. It was of interest that advancing age remained associated despite adjustment for measures of adiposity. Since adiposity is a major determinant of insulin resistance and all women had a high BMI, the association with age may be indicative of an additional mechanism such as a reduced capacity of the beta-cell to adapt to pregnancy, or a reduction in beta-cell mass or turnover, with age, or slightly lower muscle mass [22, 23]. Previous GDM remained significant after adjustment for development of GDM in the index pregnancy. The low prevalence of previous GDM in this cohort of women with obesity (3.7%) requires mention. Given that over 25% later developed GDM by IADPSG criteria, this was lower than anticipated, and may reflect use of less stringent OGTT thresholds in the previous pregnancy, lack of universal screening or a substantial inter-pregnancy increase in BMI. GDM was also associated with the presence of a first degree relative with T2DM in unadjusted analysis however the loss of this relationship following adjustment with anthropometric measures suggests that this may be underpinned by a common predisposition to obesity in family members. Other potentially relevant family histories

(GDM, ischaemic heart disease or hypertension) were not associated with development of GDM.

In contrast to national clinical guidelines [11] no evidence of a strong association between ethnic groups and development of GDM was found in this ethnically mixed cohort of women with obesity. As differences in GDM prevalence in Asian women has been attributed to variation in body fat distribution, this lack of association may reflect a lesser importance of adipose distribution when all included women are obese. A limitation is that low numbers of Asian women may have led to inadequate power to demonstrate ethnic differences.

Risk of GDM was not associated with parity, except for those who had previous GDM. This contrasts with studies in weight heterogeneous women [24, 25], and may infer that the risk associated with parity shown in other cohorts is a facet of obesity rather than parity *per se* as weight typically increases from one pregnancy to the next. The lack of association of GDM risk with socio economic status despite a positive association reported previously [26, 27] may relate to the predominant influence of BMI over socio economic status. The use of the IMD as a proxy measure of socio-economic status may also contribute.

The relationship between systolic and diastolic blood pressure and GDM in women with obesity concurs with previous descriptions in normal weight and overweight women, and in women with obesity [5, 28]. This increasing risk applied even to differences in blood pressure within the normal range. The observation that this relationship remained significant after adjustments, particularly adiposity and age, argues against the explanation of blood pressure simply being a component of the metabolic syndrome and suggests an additional mechanism such as haemodynamic influences or the impact of dietary aspects such as salt intake.

Importantly this study suggests that anthropometric measures are practically more useful in prediction of GDM than BMI in women with obesity. Most maternal anthropometric measures, including circumferences and skin fold thicknesses increased from the early second trimester to the third trimester as is known to occur in this, the anabolic phase, of normal pregnancy [29–32]. The positive difference between GDM and non-GDM women in the numerous anthropometric measures at both time points has significant mechanistic implication as it is likely to reflect different patterns of adipose distribution associated with GDM. Consideration should therefore be made as to how these anthropometric variables relate to adipose distribution and disease risk. Maternal limb and trunk circumferences are used as proxy measures for patterns of adipose distribution; in non-pregnant individuals, the waist circumference reflects a combination of central visceral adiposity and subcutaneous adipose tissue, and is an important diagnostic component of the metabolic syndrome [33]. The relationship between waist circumference and GDM in the women at the early third trimester visit, despite potential confounding by the growing conceptus is likely to reflect insulin resistance second to visceral or ectopic adiposity, a suggestion supported in a weight heterogeneous cohort [34]. Hip circumference increases in pregnancy, representing gestational accumulation of lower body fat stores; the lack of association with GDM may reflect this role. The relationship between neck circumference and GDM in women with obesity reported initially in this cohort [35] has been confirmed in an independent cohort of pregnant women with obesity [36]. Neck circumference reflects upper body subcutaneous fat which is proposed to be a unique pathogenic fat depot mechanistically related to excess non esterified fatty acids (NEFA), and has been associated with cardiometabolic risk factors after adjustment for visceral adiposity and BMI in adults, and with cardio-metabolic risk factors in children, as well as in non-obese pregnant women [37–40].

The relationship between mid-arm circumference and GDM may reflect a simple translatable surrogate for BMI measurement in pregnancy [41] and in non-pregnant individuals is associated with insulin resistance [42]. Neither wrist circumference, which has been positively

correlated with T2DM in non-pregnant women with obesity [43], nor thigh circumference, previously negatively linked with cardiovascular risk factors in non-pregnant women with obesity [44], were associated with the development of GDM in this cohort.

Other measures of body habitus (weight, BMI, waist:height, waist:hip, waist:thigh and neck: thigh ratios) were associated with GDM in unadjusted analyses, and notably waist:height ratio remained associated in the adjusted models. Whereas in non-pregnant individuals waist:height ratio, reflecting central adiposity, is known to be superior to BMI and waist circumference alone for detecting increased cardiometabolic risk [20], to our knowledge this relationship has only previously been described in pregnancy in a small group of weight heterogeneous Aboriginal women in Australia [45]. Neck:thigh ratio, reflecting upper to lower body adipose distribution, was also found to be associated with GDM after adjustment. This rarely utilised ratio theoretically represents pathological subcutaneous (and possibly visceral) adipose tissue distribution; neck circumference previously has been positively, and thigh circumference negatively associated (in women) with cardiovascular risk factors [21, 44]. An increasing ratio therefore would represent increasing cardiovascular risk. This measure, unaffected by the growing products of conception, relatively accessible and thus translatable, could prove useful, alone or in combination with other factors, for assessing GDM risk in pregnant women with obesity.

This study adds to the limited literature that skinfold thicknesses could act as potentially useful measures in GDM. Measured early in pregnancy, these provided granularity for assessing GDM risk amongst women with obesity. Skinfold thicknesses measure subcutaneous fat in a peripheral (e.g. triceps and biceps) and truncal distribution (e.g. subscapular and suprailiac) and have previously been cited as predictors for abnormal glucose and insulin regulation in non-pregnant individuals [46]. Skinfolds have most frequently been explored in pregnancy in high risk groups; for example, sum of skinfolds (triceps, subscapular and suprailiac) were associated with development of GDM particularly in South Asian women, and biceps and triceps measures were identified as predictors of hyperglycaemia when measured in early pregnancy in a cohort of predominantly black overweight women [47, 48].

Each maternal measure associated with GDM in early pregnancy (visit 1) was also associated with GDM at visit 2. Despite change in adiposity and gestational weight gain being widely considered as increasing risk for GDM in weight heterogeneous women [15, 49], there was a lack of evidence that change across gestation influences the risk of developing GDM in this cohort of women with obesity.

To our knowledge this is the most comprehensive study to have assessed maternal characteristics and detailed anthropometry longitudinally in a large cohort of pregnant women with obesity for GDM risk assessment. Limitations include the measurements being carried out in the early second trimester of pregnancy (> 15 weeks') which may limit relevance to earlier gestations. In women with obesity gestational weight and skinfold thicknesses are known to change more slowly than in normal weight or underweight women [31], and measurement at this time point may have led to an underestimation. A second limitation may lie in the use of complete-case data, however when analyses were compared with imputed data, no significant differences were found. We are cognisant that an RCT cohort was utilised to undertake this analysis with the associated potential for selection bias. A novel aspect of this study was that it was purposefully limited to pregnant women with obesity as data in this area is limited; future work might be undertaken to assess how adipose measures are translatable to normal weight women, and across ethnic groups.

There are numerous ways of selecting predictors for a single outcome from multiple correlated variables. In this study our choice was predicated on the physiological role of known predictors combined with statistical significance. Before analysis, variables were assigned to groups according to the potential for correlation (S1 Table in S1 File). For each group of

markers, after assessing correlation, the variable with the strongest association was chosen as representative of the group, in a process similar to stepwise regression. We then analysed the selected group of factors together in a single model. Other members of the groups might have performed similarly well if chosen.

In conclusion, this study has expanded knowledge of the risk factors for GDM specific to women with obesity. Classical risk factors, such as age and previous GDM were of equal importance to normal weight women. The novelty of this study lies in the observation that specific patterns of adiposity in women with obesity provide a discriminatory approach to GDM risk. Finally, weight gain or change in adiposity measures were unimportant in GDM risk, suggesting once again that most focus must be best placed on reducing adiposity prior to conception, rather than on reducing weight gain or adiposity during pregnancy in women with obesity.

## Supporting information

**S1 File.**
(DOCX)

## Acknowledgments

We thank the UPBEAT participants for their time, interest and patience, and staff and members of the UPBEAT Consortium (full list of personnel below).

UPBEAT Consortium personnel

King's College London/Guy's and St Thomas' NHS Foundation: Trust Lucilla Poston, lead author for Consortium (lucilla.poston@kcl.ac.uk), Andrew Shennan, Annette Briley, Claire Singh, Paul Seed, Jane Sandall, Thomas Sanders, Nashita Patel, Angela Flynn, Shirlene Badger, Suzanne Barr, Bridget Holmes, Louise Goff, Clare Hunt, Judy Filmer, Jeni Fetherstone, Laura Scholtz, Hayley Tarft, Anna Lucas, Tsigerada Tekletdadik, Deborah Ricketts, Carolyn Gill, Alex Seroge Ignatian, Catherine Boylen, Funso Adegoke, Elodie Lawley, James Butler, Rahat Maitland, Matias Vieira, Dharmintra Pasupathy.

King's College Hospital Eugene Oteng-Ntim, Nina Khazaezadeh, Jill Demilew, Sile O'Connor, Yvonne Evans, Susan O'Donnell, Ari de la Llera, Georgina Gutzwiller, Linda Hagg.

Newcastle University/Newcastle NHS Foundation Trust: Stephen Robson, Ruth Bell, Louise Hayes, Tarja Kinnunen, Catherine McParlin, Nicola Miller, Alison Kimber, Jill Riches, Carly Allen, Claire Boag, Fiona Campbell, Andrea Fenn, Sarah Ritson, Alison Rennie, Robin Durkin, Gayle Gills, Roger Carr.

Glasgow University and Greater Clyde Health Board Scott Nelson, Naveed Sattar, Therese McSorley, Hilary Alba, Kirsteen Paterson, Janet Johnston, Suzanne Clements, Maxine Fernon, Savannah Bett, Laura Rooney, Sinead Miller, Paul Welsh, Lynn Cherry.

Central Manchester Hospitals Foundation Trust: Melissa Whitworth, Natalie Patterson, Sarah Lee, Rachel Grimshaw, Christine Hughes, Jay Brown.

City Hospital Sunderland: Kim Hinshaw, Gillian Campbell, Joanne Knight.

Bradford Royal Infirmary: Diane Farrar, Vicky Jones, Gillian Butterfield, Jennifer Syson, Jennifer Eadle, Dawn Wood, Merane Todd.

St George's NHS Trust, London: Asma Khalil, Deborah Brown, Paola Fernandez, Emma Cousins, Melody Smith.

University College London: Jane Wardle, Helen Croker, Laura Broomfield (Weight Concern—Registered Charity.No. 1059686©)

University of Southampton: Keith Godfrey, Sian Robinson, Sarah Canadine, Lynne Greenwood.

Trial Steering Committee: Catherine Nelson-Piercy, Stephanie Amiel, Gail Goldberg, Daghni Rajasingham, Penny Jackson, Sara Kenyon, Patrick Catalano.

## Author Contributions

**Conceptualization:** Sara L. White, Dharmintra Pasupathy, Naveed Sattar, Scott M. Nelson, Lucilla Poston.

**Data curation:** Sara L. White, Shahina Begum, Paul Seed.

**Formal analysis:** Sara L. White, Shahina Begum, Paul Seed.

**Funding acquisition:** Sara L. White, Naveed Sattar, Scott M. Nelson, Lucilla Poston.

**Supervision:** Dharmintra Pasupathy, Lucilla Poston.

**Writing – original draft:** Sara L. White, Lucilla Poston.

**Writing – review & editing:** Sara L. White, Dharmintra Pasupathy, Shahina Begum, Naveed Sattar, Scott M. Nelson, Paul Seed, Lucilla Poston.

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
