## [Decision Letter · Decision Letter 0]

26 Aug 2022

PONE-D-22-18267Gestational Diabetes in Women with Obesity; an analysis of Clinical History and Simple Clinical/Anthropometric MeasuresPLOS ONE

Dear Dr. White,

Thank you for submitting your manuscript to PLOS ONE. After careful consideration, we feel that it has merit but does not fully meet PLOS ONE’s publication criteria as it currently stands. Therefore, we invite you to submit a revised version of the manuscript that addresses the points raised during the review process.

We look forward to receiving your revised manuscript.

Kind regards,

Callam Davidson

Editorial Office

PLOS ONE

Journal Requirements:

"SLW, DP, SB, SMN, PS and LP none.

I have read the journal's policy and NS has consulted for Abbott Diagnostics, Amgen, AstraZeneca, Boehringer Ingelheim, Eli Lilly, MSD, Novo Nordisk, Pfizer, and Sanofi and his University has received funding from grants from Astrazeneca, Boehringer Ingelheim, Roche diagnostics."

Additional Editor Comments:

Please ensure that the study is reported according to the STROBE guideline, and include the completed STROBE checklist as Supporting Information. Please add the following statement, or similar, to the Methods: \\"This study is reported as per the Strengthening the Reporting of Observational Studies in Epidemiology (STROBE) guideline (S1 Checklist).\\"\\n\\nThe STROBE guideline can be found here: http://www.equator-network.org/reporting-guidelines/strobe/\\n\\nWhen completing the checklist, please use section and paragraph numbers, rather than page numbers."}" style="color: rgb(0, 0, 0); font-size: 10pt; font-family: Arial;"> Please ensure that the study is reported according to the STROBE guideline, and include the completed STROBE checklist as Supporting Information. Please add the following statement, or similar, to the Methods: "This study is reported as per the Strengthening the Reporting of Observational Studies in Epidemiology (STROBE) guideline (S1 Checklist)."

Reviewers' comments:

Reviewer's Responses to Questions

**Comments to the Author**

1. Is the manuscript technically sound, and do the data support the conclusions?

Reviewer #1: Yes

Reviewer #2: Yes

2. Has the statistical analysis been performed appropriately and rigorously? 

Reviewer #1: Yes

Reviewer #2: Yes

3. Have the authors made all data underlying the findings in their manuscript fully available?

Reviewer #1: No

Reviewer #2: Yes

4. Is the manuscript presented in an intelligible fashion and written in standard English?

Reviewer #1: Yes

Reviewer #2: Yes

5. Review Comments to the Author

Reviewer #1: Authors analyzed 1117 women with obesity to investigate the association of the development of GDM with clinical risk factors and anthropometric measurements. The results showed that age, previous GDM, systolic blood pressure, second trimester, subscapular skinfold thickness and neck circumference were associated with later development of GDM.

1. Line 79. Women with hypertension were excluded. Please clarify the reason for the exclusion criteria. Will this influence the generalization of results from current study?

2. Line 84. Women were randomized to intervention and control group in the UPBEAT trial. Will different treatments matter in the final results reported in this study?

3. Line 96. Please provide more detail information on the imputation strategy.

4. Line 98. Please provide the definition of outliers here.

5. Line 139. “the factor with the strongest association…. Selected for inclusion.” As the model included more than one variable, please clarify this.

6. Line 160. “measure falling >=4SD from the mean” what if the skewed data? Or any skewed variables in the analysis?

Reviewer #2: The authors present the results of a secondary analysis of the UPBEAT interventional trial. This secondary analysis aimed to examine the association of maternal clinical and demographic factors and anthropometric measures, assessed in the early second and third trimester, with the development of GDM. The authors confirmed the association of classic risk factors with the development of GDM and added information on the risk of GDM that is related to certain adiposity indices. The paper is very well presented and importantly, doesn’t over interpret their findings.

1. General : Overall well written but some sentences are long and hard to follow (example introduction line 68-72)

2. Small issue but in the original RCT 1555 were randomized but in this paper (results line 165) it is stated that there were 1554. Typo?

3. As this is a cohort study, I feel that it would be worthwhile mentioning how the RCT cohort was developed and specifically that 6704 (76%) of the eligible women declined participation. Speaks to potential selection bias and generalizability.

4. Table 1: I would suggest adding Upbeat study group assignment as this would be considered a “baseline” characteristic

5. Was there consideration of Bonferroni correction for multiple comparisons?

6. Tables 3 and 4 – the explanation of the meaning of “/” and “`” in the footnote is confusing as a part of one continuous sentence. Please give each symbol its own line.

7. I wonder how the measures of adiposity would be represented in a non obese group. Any data from the literature? This is relevant as SC adiposity and/or visceral adiposity can be increased without leading to a BMI defined obesity. Consider addressing in the discussion as this would provide context for implications in all pregnancies (future research?).

8. The authors appropriately provide information on the correlation of the different adiposity measurements with visceral vs superficial adiposity in the discussion. I wonder whether there should be some explanation of this in the introduction to explain the rationale for the use of the chosen anthropometric measurements in this study.

9. Discussion line 350 – grammar - plural vs singular.

6. PLOS authors have the option to publish the peer review history of their article (what does this mean?). If published, this will include your full peer review and any attached files.

Reviewer #1: No

Reviewer #2: **Yes: **Howard Berger

---

## [Author Response · Author response to Decision Letter 0]

27 Oct 2022

Dear reviewers and editorial team,

Thank you very much for reviewing this manuscript and for the comments provided. Please find our point-by-point response below.

Reviewer #1: 

1. Line 79. Women with hypertension were excluded. Please clarify the reason for the exclusion criteria. Will this influence the generalization of results from current study?

Women with pre-existing hypertension were not randomised to the original study, however no women were excluded if they subsequently developed hypertension during pregnancy, a much more common scenario in this group of women. 

2. Line 84. Women were randomized to intervention and control group in the UPBEAT trial. Will different treatments matter in the final results reported in this study?

We note that there was no impact of the intervention on the primary outcomes, but cannot completely exclude some effect of the intervention, however we have minimised any impact by adjusting for randomisation arm in any relevant analysis (noted in section 2.4: selection of confounders).

3. Line 96. Please provide more detail information on the imputation strategy.

We have added further detail on the imputation strategy into the text as requested (Section 2.1)

4. Line 98. Please provide the definition of outliers here.

Thank you – we have now added this detail here.

5. Line 139. “the factor with the strongest association…. Selected for inclusion.” As the model included more than one variable, please clarify this.

Thank you for this astute observation. Each variable from the group was tested for its association individually with GDM and the variable with the highest z score selected for inclusion. I have updated the text to reflect this.

6. Line 160. “measure falling >=4SD from the mean” what if the skewed data? Or any skewed variables in the analysis?

The only variable noted as significantly skewed was BMI and it was not included in the identification of outliers.

Reviewer #2

1. General : Overall well written but some sentences are long and hard to follow (example introduction line 68-72)

Thank you. We agree and have simplified the above paragraph.

2. Small issue but in the original RCT 1555 were randomized but in this paper (results line 165) it is stated that there were 1554. Typo?

Thank you – I have corrected the text. (This was not actually a typo - the number of women randomised was indeed 1555, however one woman was subsequently excluded from the standard arm as she was recruited concurrently into a different trial as shown in the original trial flow diagram (https://www.thelancet.com/journals/landia/article/PIIS2213-8587(15)00227-2/fulltext#figures – however to avoid confusion, we have corrected to the original total RCT figure).

3. As this is a cohort study, I feel that it would be worthwhile mentioning how the RCT cohort was developed and specifically that 6704 (76%) of the eligible women declined participation. Speaks to potential selection bias and generalizability.

Thank you for this suggestion. On balance, although we agree that it does speak to potential selection bias, we feel that this level of numerical detail might be excessive for this manuscript. We have however added a sentence to the limitations section to this effect (L384).

4. Table 1: I would suggest adding Upbeat study group assignment as this would be considered a “baseline” characteristic

 Thank you – we have added this into Table 1.

5. Was there consideration of Bonferroni correction for multiple comparisons?

During the planning stages of this analysis careful consideration was given to the variables that would be tested, and only those felt to be ‘a priori’ associated with GDM, out of a much larger dataset, were chosen for analysis. As a result of this, the team, including the statistician, agreed that Bonferroni correction for multiple comparisons was not necessary.

6. Tables 3 and 4 – the explanation of the meaning of “/” and “`” in the footnote is confusing as a part of one continuous sentence. Please give each symbol its own line.

Thank you – this has been rectified.

7. I wonder how the measures of adiposity would be represented in a non obese group. Any data from the literature? This is relevant as SC adiposity and/or visceral adiposity can be increased without leading to a BMI defined obesity. Consider addressing in the discussion as this would provide context for implications in all pregnancies (future research?).

Thank you for this observation; there is reference to some measures in normal and weight heterogeneous women in the text (refs 34, 40, 45, 47) however there is room for further exploration and we have added a sentence reflecting this to the discussion.

8. The authors appropriately provide information on the correlation of the different adiposity measurements with visceral vs superficial adiposity in the discussion. I wonder whether there should be some explanation of this in the introduction to explain the rationale for the use of the chosen anthropometric measurements in this study.

Thank you – we have added an explanatory phrase in the Section 2.2 maternal clinical factors

9. Discussion line 350 – grammar - plural vs singular. Thank you – rectified.

In response to your prompt with regards to data sharing, we recommend the following text:

“Due to the limitations of the consent provided by the patients in our study, and restrictions imposed by our funders we cannot make the data generally available. The UPBEAT Scientific Advisory Committee accept applications for use of data from those who make a formal request, providing a description of the intended study on a research application form (UPBEAT RAF) available from Glen Nishku (glen.nishku@gstt.nhs.uk). Providing the proposed studies do not conflict with consent, the data will be freely available.”

With regards to Point 4 regarding a data repository, as explained in the data sharing paragraph above the data will only be available on request.

Finally, here is our updated competing interests statement:

“SLW, DP, SB, SMN, PS and LP none.

I have read the journal's policy and NS has consulted for Abbott Diagnostics, Amgen, AstraZeneca, Boehringer Ingelheim, Eli Lilly, MSD, Novo Nordisk, Pfizer, and Sanofi and his University has received funding from grants from Astrazeneca, Boehringer Ingelheim, Roche diagnostics. This does not alter our adherence to PLOS ONE policies on sharing data and materials.”

With best wishes and thank you very much for your patience,

Dr Sara L White MRCP MSc FRCPath EuSpLM PhD

Clinician Scientist and Maternal Diabetes Clinical Research Lead, 

Department of Women and Children’s Health, King’s College London

Honorary Consultant in Metabolic Medicine (Clinical Biochemistry), Guy’s and St Thomas’ NHS Foundation Trust

---

## [Decision Letter · Decision Letter 1]

14 Nov 2022

PONE-D-22-18267R1Gestational Diabetes in Women with Obesity; an analysis of Clinical History and Simple Clinical/Anthropometric MeasuresPLOS ONE

Dear Dr. White,

Thank you for submitting your manuscript to PLOS ONE. After careful consideration, we feel that it has merit but does not fully meet PLOS ONE’s publication criteria as it currently stands. Therefore, we invite you to submit a revised version of the manuscript that addresses the points raised during the review process.

The comments of reviewer 1 have been satisfied but reviewer 2 has raised queries that need to be addressed including modification of your overall conclusion and addressing the Strobe requirements

We look forward to receiving your revised manuscript.

Kind regards,

Stephen L Atkin, MD

Academic Editor

PLOS ONE

Reviewers' comments:

Reviewer's Responses to Questions

**Comments to the Author**

1. If the authors have adequately addressed your comments raised in a previous round of review and you feel that this manuscript is now acceptable for publication, you may indicate that here to bypass the “Comments to the Author” section, enter your conflict of interest statement in the “Confidential to Editor” section, and submit your "Accept" recommendation.

Reviewer #1: All comments have been addressed

Reviewer #2: (No Response)

2. Is the manuscript technically sound, and do the data support the conclusions?

Reviewer #1: Yes

Reviewer #2: Yes

3. Has the statistical analysis been performed appropriately and rigorously? 

Reviewer #1: Yes

Reviewer #2: I Don't Know

4. Have the authors made all data underlying the findings in their manuscript fully available?

Reviewer #1: (No Response)

Reviewer #2: Yes

5. Is the manuscript presented in an intelligible fashion and written in standard English?

Reviewer #1: (No Response)

Reviewer #2: Yes

6. Review Comments to the Author

Reviewer #1: (No Response)

Reviewer #2: I would like to thank the authors for their thoughtful revisions in response to the reviewers’ comments. A few residual comments, questions:

1. Abstract : conclusions. I just am not sure how “modifiable” the identified risk factors are. Age is not modifiable and blood pressure is possibly a marker of GDM risk but not a causative factor. Furthermore, I am not aware of any methods of modifying subscapular adiposity and neck thickness and whether these are again just indicators of baseline increased insulin resistance. I suggest modifying the conclusions in the abstract so that they are more in keeping with the correctly worded (in my opinion) conclusions in the discussion.

2. Methods, Page 5, line 108: Not sure what 27+8 weeks means. Is it 28+1?

3. If the GTT was performed at gestational ages beyond the accepted window of 24-28 weeks of gestation (as mentioned in the methods line 106-108) then I suggest adding the GA at GTT as a demographic variable in Table 1. If there is a difference in mean GA at GTT between GDM and non GDM then this variable would need to be added to the logistic regression models.

4. Thank you for your explanation regarding the reasoning behind not presenting the base population from which the cohort was derived which is actually the base population that created the UPBEAT trial study population. I still think that if adhering to the STROBE guidelines (as indicated by the authors) one needs to present the results starting with a figure showing the cohort development, starting with the 8820 women screened for UPBEAT. I will leave it to the editor to decide whether this is correct.

7. PLOS authors have the option to publish the peer review history of their article (what does this mean?). If published, this will include your full peer review and any attached files.

Reviewer #1: No

Reviewer #2: No

---

## [Author Response · Author response to Decision Letter 1]

15 Nov 2022

Dear Professor Atkin,

Thank you very much for reviewing our updated manuscript and for the comments provided. Please find our point-by-point response below.

Reviewer #2: 

1. Abstract : conclusions. I just am not sure how “modifiable” the identified risk factors are. Age is not modifiable and blood pressure is possibly a marker of GDM risk but not a causative factor. Furthermore, I am not aware of any methods of modifying subscapular adiposity and neck thickness and whether these are again just indicators of baseline increased insulin resistance. I suggest modifying the conclusions in the abstract so that they are more in keeping with the correctly worded (in my opinion) conclusions in the discussion.

Thank you for identifying this ambiguity in the abstract – we have now updated the abstract to reflect the discussion conclusion more closely.

2. Methods, Page 5, line 108: Not sure what 27+8 weeks means. Is it 28+1?

Apologies, this was an error – it should have read 27+5, now corrected.

3. If the GTT was performed at gestational ages beyond the accepted window of 24-28 weeks of gestation (as mentioned in the methods line 106-108) then I suggest adding the GA at GTT as a demographic variable in Table 1. If there is a difference in mean GA at GTT between GDM and non GDM then this variable would need to be added to the logistic regression models.

Gestational age at visit 2 (OGTT) is included in Table 3 as the timepoint of these data measures; it does not differ between GDM and non GDM groups. We have added a note to this variable in Table 3 to improve clarity. 

 4. Thank you for your explanation regarding the reasoning behind not presenting the base population from which the cohort was derived which is actually the base population that created the UPBEAT trial study population. I still think that if adhering to the STROBE guidelines (as indicated by the authors) one needs to present the results starting with a figure showing the cohort development, starting with the 8820 women screened for UPBEAT. I will leave it to the editor to decide whether this is correct.

We appreciate your request and have added a phrase into the Study Design noting the total number of women approached for inclusion. 

Editor comment:

The comments of reviewer 1 have been satisfied but reviewer 2 has raised queries that need to be addressed including modification of your overall conclusion and addressing the Strobe requirements

We hope that we have now responded adequately to these queries.

With best wishes and thank you once again for considering our manuscript,

Dr Sara L White MRCP MSc FRCPath EuSpLM PhD

---

## [Decision Letter · Decision Letter 2]

12 Dec 2022

Gestational Diabetes in Women with Obesity; an analysis of Clinical History and Simple Clinical/Anthropometric Measures

PONE-D-22-18267R2

Dear Dr. White,

We’re pleased to inform you that your manuscript has been judged scientifically suitable for publication and will be formally accepted for publication once it meets all outstanding technical requirements.

Kind regards,

Stephen L Atkin, MD

Academic Editor

PLOS ONE

Additional Editor Comments (optional):

Reviewers' comments:

Reviewer's Responses to Questions

**Comments to the Author**

1. If the authors have adequately addressed your comments raised in a previous round of review and you feel that this manuscript is now acceptable for publication, you may indicate that here to bypass the “Comments to the Author” section, enter your conflict of interest statement in the “Confidential to Editor” section, and submit your "Accept" recommendation.

Reviewer #2: All comments have been addressed

2. Is the manuscript technically sound, and do the data support the conclusions?

Reviewer #2: (No Response)

3. Has the statistical analysis been performed appropriately and rigorously? 

Reviewer #2: (No Response)

4. Have the authors made all data underlying the findings in their manuscript fully available?

Reviewer #2: (No Response)

5. Is the manuscript presented in an intelligible fashion and written in standard English?

Reviewer #2: (No Response)

6. Review Comments to the Author

Reviewer #2: (No Response)

7. PLOS authors have the option to publish the peer review history of their article (what does this mean?). If published, this will include your full peer review and any attached files.

Reviewer #2: No

---

## [Editor Report · Acceptance letter]

21 Dec 2022

PONE-D-22-18267R2 

Gestational diabetes in women with obesity; an analysis of clinical history and simple clinical/anthropometric measures 

Dear Dr. White:

I'm pleased to inform you that your manuscript has been deemed suitable for publication in PLOS ONE. Congratulations! Your manuscript is now with our production department. 

Kind regards, 

on behalf of

Dr. Stephen L Atkin 

Academic Editor

PLOS ONE